# Mary of Bethany: Creation through Conversation

**Susan Fish**

Theological Studies, Conrad Grebel University College, University of Waterloo, Waterloo, ON N2L 3G6, Canada; susan@storywell.ca

**Abstract:** The author uses the story of Mary of Bethany anointing Jesus's feet in John 12 as a jumping-off point for considering the prophetic role of artistic conversation, in the Gospel of John, in the whole Bible and in her own artistic life.

**Keywords:** Mary of Bethany; artistic conversation; performance art

## 1. Introduction

Visual artist and theologian Makoto Fujimura says that "artists can open new doors of theological illumination" (Fujimura 2020, p. 4) precisely because they move away from propositional and analytical ways of thinking and into a more generative path. Such a path, however, is not that of a lone artist working in isolation, but rather in conversation.

Further, we know that art is not only a conversation between artist and audience, but also between artists whose creativity and fresh exvpressions are sparked by those of other artists. Similarly, theology is both a conversation between people building on one another's ideas about God and giving them new forms. At its best, a conversation can be both theological and artistic, where the very form of conversation can be said to imitate perichoresis, the dancing relationship of the Trinity, with its ripple effects continuing to move outward and into imaginations and lives. Such conversations are most theologically creative when audience members are themselves transformed into artists.

In this paper, I seek to demonstrate this kind of conversation between artists for theological purposes in the Gospel of John. I explore an exemplar of such a conversation at the turning point between the first and second halves of the Gospel, in the pericope of the anointing of Jesus by Mary of Bethany in John 12. While this scene might seem to merely describe a simple gesture by a sister grateful for the restored life of her brother, I argue instead that the Gospel writer found in Mary of Bethany a kindred spirit whose actions can be seen as performance art that is both theologically prophetic and artistically analogous to what the evangelist is seeking to do in the writing of the Gospel. Both the Gospel writer's and Mary's artistry in John 12:1–8 occur in conversation with Jesus's acts throughout the Gospel, and in contrast with Judas, who engages in his own inverted version of performance art in this pericope. Further, it is possible that even Jesus's own subsequent actions may be influenced by Mary's, suggesting that such art has a contagious or ripple effect. In the second half of this paper, as a working artist myself—a novelist—I demonstrate how this process continues to occur in conversations among artists with one of its purposes being the ongoing effects in the creative lives of those encountering the art.

## 2. The Artistry of the Fourth Gospel

The Gospel of John has long been recognized as a gospel "composed by a very creative and highly capable author" (Luther 2020) who intentionally borrows from storytelling techniques and devices of ancient dramas for rhetorical effect. Among those who explore this dimension are Jo-Ann Brant in *Dialogue and Drama: Elements of Greek Tragedy in the Fourth Gospel* (2004) and, more recently, George L. Parsenios and Harold Attridge. Cornelia van Deventer, whose work focuses on the literary features and rhetorical effects of the

Gospel of John, draws on biblical performance criticism as she examines the power of performance not only to evoke meaning but also to "work on or among the group of hearers in the context" (van Deventer 2019, p. 519). Van Deventer notes that the implied audience for the Gospel is spelled out plainly in John 20:30—31 and in the prologue, which gives the audience a privileged understanding of what they are about to see and hear so that they become insiders (Ibid., pp. 519–20). Brant says that the descriptive scenes in John serve to turn the readers of the Fourth Gospel into eyewitnesses themselves (Brant 2004, p. 41) cited in (van Deventer 2019, 522 footnotes). Those eyewitnesses then can in turn, and in their own creative ways, bear witness to what they have experienced in ever-widening circles.

Among the techniques used to achieve this rhetorical effect is the highly sensory language of John. In her essay "The Gospel of John and the Five Senses", Dorothy Lee says the reader must use their senses to enter imaginatively into the highly symbolic Johannine faith with its focus on embodiment (Lee 2010, p. 116). The key to this reading is found in the opening words of John's first epistle—"That which was from the beginning, which we have heard, which we have seen with our own eyes, which we have beheld and our hands have handled" (I John 1:1)—pointing to the sensuous approach of the writing of this Gospel writer. New Testament scholar Jaime Clark-Soles draws on scholarship around embodiment, sensory studies, and participatory theology to show how the sensory details found in this Gospel serve as an entry point for experiencing God in the Gospel of John. While each of the Gospels presents a story of Jesus being anointed by a woman (who is only sometimes named, with the stories possibly referring to different events), only the writer of John mentions the aroma of the perfume filling the whole house, likely because in its original context, it was not necessary to spell this out: all oils were aromatic (Kurek-Chomycz 2010, p. 337). But, says New Testament scholar Dominika A. Kurek-Chomycz, "this is precisely why we need to understand it as more than just a piece of factual information, supplied allegedly by an eye (nose?)-witness" (Ibid.) for readers who enter the story with the help of such details and become witnesses themselves.

Another technique for achieving this effect is the dichotomous language found within the Fourth Gospel. Van Deventer says that throughout the Gospel, "characters express views and model behaviours that identify them either as part of those who accept Jesus and become children of the divine . . . or as part of those who reject him"(van Deventer 2019, p. 524) When it comes to characters within the Gospel (and also, by rhetorical implication, the audience), as Christopher W. Skinner notes (Skinner 2016, p. 124, quoted in van Deventer 2019, p. 524), there is no middle ground. Again, the rhetorical purpose of these presentations is to encourage the audience to become "participators and propagators" (Loubser 2013, p. 169, quoted in van Deventer 2019, p. 530) of Jesus's life and words. In other words, the artistry of the writer and the artistic subjects of scenes like that in the John 12 pericope invite the reader into their own faithful artistic expression.

### 3. The Artistry of Mary of Bethany (John 12:1–8)

Let us turn to the pericope. Chapter 12 of the Gospel of John opens on a new scene, but one which must be seen in the context of other scenes in the Gospel. The indication that this is so comes from chapter 11, in which the story of the raising of Lazarus identifies Mary proleptically as the one who (will) anoint Jesus's feet and dry them with her hair, while in John 12 there are two reminders that Lazarus is the man Jesus raised from the dead. Further, the supper in this scene seems to celebrate Lazarus being restored to life, given that Mary, Martha, and Lazarus are named as present. Within the sweep of the Gospel, this scene is a turning point, marking the end of Jesus's ministry and the turn toward death (Kurek-Chomycz 2010, p. 339). This is fitting because the celebratory feast parallels the meal that came at the start of Jesus's ministry: the wedding at Cana (Ibid., p. 337). A further parallel can be seen between the good wine Jesus produces at Cana and the expensive ointment Mary pours on Jesus's feet (Brant 2013, p. 89).

On the surface, Mary's action is simply a "gesture of gratitude" (Brant 2013, p. 90) for the raising of her brother, an act by a woman whom feminist interpreters have critiqued as silent and more passive than her sister who is praised for "Martha's forthright faith and active orientation" (Beavis 2012, p. 741). Mary Ann Beavis explores various feminist points of view on Mary's silence (particularly whether her silence reflects Jesus' restriction of women or the writer's rhetorical interests) but regardless of the reason, while Mary of Bethany is seemingly a woman of means, she is still a woman and thus a silent disciple.

Yet, in this scene in the Fourth Gospel, while Mary is silent, she is anything but passive. In fact, she may be engaging artistically in extraordinary prophetic theology not with words but with scent as her artistic medium and her contribution to the theological conversation. In his examination "Religion and Art Theater", Bryan Rennie points out, "the more rigorous the life the more creative the one living it must become" (Rennie 2015, p. 331). Rennie looks at the Hebrew prophets during times of captivity or foreign occupation:

> . . .the Hebrew prophets were "performance artists" who used poetry and the verbal arts as well as dramatic performance in order to accomplish their role as diviners, communicating their own emotional responses to abducted agency in their environment, commending specific behavior by commanding the attention of their audiences through the artful expression of that response." (Ibid., p. 329)

Similarly, in the John 12 pericope, Mary's silence reflects the abducted agency of female disciples, but, like the prophets, she speaks without words through her artful act of anointing Jesus.

The fact that her actions are prophetic is pointed out by Susan Miller, who notes the unusual elements of Mary's actions: Mary anoints Jesus during the meal rather than beforehand; she anoints Jesus' feet rather than his head; she loosens her hair in public—a shocking and potentially shameful act; and removes the perfume by drying Jesus's feet. The unusualness of each of these actions "indicate that her gift may be interpreted as a symbolic action reminiscent of the actions of prophets" (Miller 2007, p. 240) such as Ezekiel or Jeremiah whose actions embodied their prophetic message. As a woman, Mary may be consigned to a silent role, but her public gesture of anointing Jesus enables her to communicate an important message without saying a word (Ibid.).

In the choice of nard as the anointing oil, she evokes the Song of Songs, the only place nard is mentioned in the Hebrew Bible (Song of Songs 1:12, 4:13, 14), with Mary taking on the role of the female lover (Beavis 2012, p. 745). While Kurek-Chomycz says the mixing of the smells of nard, Jesus's feet, and Mary's hair "is not devoid of sexual undertones" (Kurek-Chomycz 2010, p. 343), the pericope points in a different direction so that it is a recognition of Jesus as messiah and king, as he, like the king in Song of Songs, reclines at a banquet table (Ibid., p. 342).

But it goes further: anthropologist David Howes observes the universal association between smells and times of transitions, most particularly in the rituals and acts connected with birth and death (Howes 1991, quoted in Kurek-Chomycz 2010, p. 339). At one level, this is the Gospel writer's artistry—situating the story at this point in the narrative—but it also points to what Mary is doing which, according to Jesus, is anointing his body in preparation for death (Mark 14:8), an act that Kurek-Chomycz says implies Mary's "foreknowledge, or rather forescent of the impending events" (Ibid., p. 341). Of this prophetic element, Francis J. Moloney writes:

> For the first time in the narrative, Jesus' proximate death is recognized. Mary is the first to accept that the illness and death of Lazarus will be the means by which the Son of God will be glorified (11.4). Jesus' supportive explanation of Mary's gesture in v. 7 indicates that, at last, one of the characters in the story has "got it right." (Moloney 2003, p. 525)

That this is a unique posture can be seen in contrast with both Martha and Judas. In John 11, Martha is hesitant to open Lazarus's tomb, for fear that opening it will release a bad stench. Here, Mary is able to respond differently to death: as Kurek-Chomycz says,

"The scent signifying death-which-leads-to-resurrection is in this way juxtaposed with the stench of death to which Martha referred to in the comment on her brother's corpse" (Kurek-Chomycz 2010, p. 341). The author of John has been described as having "the ability of an artist to transform anguish into something sublime" (Brant 2013, p. 83), but here, Mary also demonstrates she can recognize the potential for beauty in the looming death of Jesus.

For Judas, by contrast, death can only stink. Here, Judas is singled out as the one person who critiques and misunderstands Mary's act. But Judas, too, is engaged in performance, pretending he has a concern for the poor he does not have. In reality, Judas is plotting to betray Jesus. For Judas, this moment in John's Gospel shows he does not understand the "death-that-leads to resurrection" and thus is attempting both to find another way, and to pretend he understands the purpose of Jesus's mission. The fact that a woman (not a priest) anoints Jesus's feet (and not his head) shows the Kingdom of God is not unfolding as Judas and perhaps others expect (Kurek-Chomycz 2010, p. 344).

Not only do Mary's actions find resonance with the Gospel writer who tells her story, but also with the actions of Jesus. As mentioned, she evokes the abundance seen in the wedding at Cana, but her actions also point to, and perhaps even give creative ideas to Jesus for how he will demonstrate his love for his disciples when he washes their feet: the very first action shown in John 13 uses the same verb for wiping feet as is used for Mary's wiping of Jesus's feet with her hair. As Clark-Soles says, "She does what Jesus commands before he commands it". Mary is thus the exemplary disciple whose opposite can be found in Judas.

## 4. Conversation among Artists

Mary's actions as conveyed by the Gospel writer are intended to have resonance with the reader, who is invited to enter into the story and to find their own life as they re-imagine and re-create the story in their own contexts. Lee says the incarnation in the Gospel of John not only reveals God but also shows "the dynamism inherent in creation itself and the possibilities of new life in the perpetual and unceasing (re)creativity of God" (Ibid., p. 127). This artistic rendering of new life is what Mary and the writer of the Fourth Gospel engage in as exemplars and call others to as disciples.

Here, I turn to my own experience of artistic conversation among disciples.

One evening last April, I was finishing the composition of a novel when I attended a concert given by the Da Capo Choir. Their director composes (and commissions) music, using poetry from around the world as their lyrics. Their second piece that evening was composed by Jeffery Van, who used Walt Whitman's Civil War poetry. The piece ended with the poem "Reconciliation" and these words:

> Word over all, beautiful as the sky!
>
> Beautiful that war and all its deeds of carnage must in time be utterly lost,
>
> That the hands of the sisters Death and Night incessantly softly wash
>
> again, and ever again, this soil'd world

What caught my attention was the equivocal meaning of the word soil'd: it can mean spoiled, but it can also mean made of good soil. In essence, it is a question about our theological orientation to the world—whether we see it as essentially fallen or good. That was already a question in my novel.

That the question even arose for me is an example of the conversations and experiences that had already infused my writing. It was spring and I had been working in my garden as well as working on this novel, which is loosely based on the life of my great grandfather who was a gardener in the Isle of Man. The narrative question of the book has to do with his vocation—rather than going down into the mines and quarries as his fathers had, my great grandfather worked in a garden. The narrative question I pursued in this book had to do with the pull of the mines and the draw of the garden.

The lyrics also took me back to a time two years before, when this novel was just a twinkle in my eye. As part of a placement for my theological studies, my husband and I moved to a rural retreat center where I was to host a variety of programs. Because of COVID-19 lockdowns, this became mostly impossible. My word of the year that year was compost, and so I composted my expectations and my learning shifted to deep conversation with the place and with the writers in my reading list.

I thought I would write fiction, but I did very little. Instead, I kept a journal in which I wrote, "I've done some writing and maybe set myself on the right path of it, but more than that I am preparing the soil. We save our vegetable peels, the eggshells from the chickens here, and we are bringing those home with us again to add to the composter, because it tells the story of where we have been. It feeds the soil that is our everyday life with the compost that has been the time here".

That was what another Mary did, she "treasured up all these things and pondered them in her heart".

That summer, I engaged in two key spiritual practices: every day, I walked the seven-circuit labyrinth at the retreat center, with field stones marking the pathways, and bushes surrounding it. The second practice was one I fell into: deciding I needed to ground myself in the place, I ventured out on my bicycle to ride hills of the six-mile square of gravel roads that surround the retreat center property.

I also made things: on one of the first days there, I was taught how to make jelly from wild violets that grew on the property, and I did so. Walking the labyrinth became a kind of performance art—one day I walked it backwards, another time I sprinkled wildflower seeds. I biked its curves (unsuccessfully) and left a coin hidden among daffodils. One evening, I put three tiny stones on the large rock at the center of the labyrinth. The next night, I thought perhaps there was one more small stone in the center. An hour later, I met our neighbor, who had just finished quarantining alone in a small retreat house. She asked if I had been the one to leave the trail of stones, said she had left the other stone. "We will talk in the language of rocks", we said.

The experience turned less idyllic when I found a tick on my skin and then when we came across a large snake on the road and I badly strained a quad muscle, leaping into the air. I also watched the director and her family and others hard at work, and recognized that I was privileged to enjoy rural life without embracing its challenges. I learned that the chickens whose eggs I collected each day would peck if I wore red, reading it as blood, as weakness.

When my snake injury meant I could neither bike nor hike for a few days, I turned to making labyrinths out of sourdough and paint on canvas. I wrote a poem about snakes and labyrinths, a shape poem, forming the shape of a labyrinth or a curled-up snake, going from the outside to the very center:

> What if a snake is a labyrinth or a labyrinth a snake? Surely I am not the first to think of it, to make the connection. Perhaps the idea was in the image from the very start. Perhaps the labyrinth is a milksnake, the one everyone reminded me was harmless, ate bugs and mice, but which I reared up from as though I was the snake, the wounding in the fear, the snake trusting and resting in the providence and leading of the one who made it, God.

I also read Wendell Berry's *Home Economics*, a collection of essays that are an elegy to slowness and smallness and his chosen agricultural life, even as I saw that the retreat center property was too big for the director and her family to manage with the skeleton staff. In his essay "Two Economies", Berry contrasts the industrial economy with the Great Economy (which he also calls the Kingdom of God). He says that, unlike wild creatures, humans "may choose not to live in [the Great Economy]—or, rather, since no creature can escape it, they may choose to *act* as if they do not, or they may choose to try to live in it on their own terms" (Berry 1987b, p. 58). As an example of the Great Economy, he describes the tending of topsoil—something that cannot be made but can be cared for. By contrast, those operating out of the industrial economy see soil as only a fund: "The invariable

mode of its relation both to nature and human culture is that of mining: withdrawal from a limited fund until that fund is exhausted" (Ibid., p. 68).

These were the exact choices being made by my character: to farm or to mine. To choose life or death. To see the world as good (although containing ticks and snakes and hard labor) or to see it as sinful.

In *Home Economics*, Berry (1987a) points briefly to Walt Whitman's poem "This Compost". In that poem, Whitman writes of Civil War graves:

Behold this compost! behold it well!

Perhaps every mite has once form'd part of a sick person—yet behold!

The grass of spring covers the prairies, . . .

What chemistry!

That the winds are really not infectious,

That this is no cheat, this transparent green-wash of the sea which is so amorous after me. . .

That all is clean forever and forever,

That the cool drink from the well tastes so good,

That blackberries are so flavorous and juicy.

The process of composting turns death to life. A conversation is a kind of compost too, the soil eternally renewing if it is tended well. That is what art is, a never-ending conversation between mediums—from poetry to music to song to fiction, from performance art to baking to memoir to theology.

Many visual artists have an artist's statement describing in words their intentions expressed in pictures or sounds. Writing often goes the other way around—because writing is already abstracted from the physical into words, anything too obviously stated becomes a moralizing sermon.

I added the words—Whitman's, interpreted by Van in 1986, and again by the choir in 2023—to a story set in the early 20th century. The book itself begins with two epigraphs, one from Carl Jung that forms my own purpose in writing and the other from Frederick Buechner that forms my theological answer:

I am under the influence of questions which were left incomplete
or unanswered by my ancestors.

Carl Jung

Here is the world.
Beautiful and terrible things will happen.
Don't be afraid.
Frederick Buechner

## 5. Artistic Compost

I close this essay with the scene I wrote after hearing the choir sing Whitman's words, and after I let Berry's words and my experience at the retreat center compost deep within me:

The light shone through the window and made squares on the floor of the church. John was watching them when he heard the preacher say, "He came to this soil'd earth".

*He* had—Jesus and John Cowin alike. Since he had been working the gardens at Foxdale House, John heard the Bible stories differently, saw that the Lord loved the earth, the land as he did. Unless a grain of wheat falls into the ground and dies, it remains alone. Consider the lilies of the field. The wheat and the tares. A sower went out to sow. A good tree does not bear bad fruit. There was a man who had planted vines to grow grapes and built a wall around the garden and went to another country, sending servants and then his son at the harvest.

John did not know how deeply the Bible passages had been worked into the soil of his mind and heart until the words rose to the surface, seeds he did not know had been planted.

*This soil'd earth.* How the Savior loved soil and growing things.

But now John roused himself from his reverie and the preacher—who, like James and John and Peter and Andrew, had been a fisherman until he took a hook to the eye and now wore a patch as he preached, so that John was not sure whether he could even see the left side of the small congregation let alone the squares of light that marked time and the length of the sermon as they crossed the altar—the preacher was repeating the word soiled. "Soiled. Spoiled. Sullied. Stained", the preacher said. "This wretched, unredeemed world. I wonder whether God is constantly tempted to send another flood to destroy every living thing".

John looked about him at the faces of miners scrubbed ruddy and at wind-burned fishermen and farmers, their sun-kissed wives and children, all of them like flowers with their faces turned toward not the sun, but the preacher who condemned them. He thought about how God had shaped Adam out of the earth, the soil, and made him in God's image.

Now, the preacher was speaking of brimstone and likening it to the smelters and smokestacks of Foxdale, saying God had come once with water but in the end would come with fire.

But God had not destroyed every living thing with the flood. He had carefully preserved life, just as Susan Kitto collected seeds from plants for the next year. God had preserved even in the face of wickedness, as John and Susan Kitto had covered the tender seedlings when the smell of the air augured frost, as John had fed his new lambs with a cloth soaked in milk.

*This soil'd earth.* Perhaps John was no preacher, but perhaps this preacher was no Jesus. This preacher talking of the laundry now, the soiled laundry, the marred creation.

John thought of the small nest he had discovered in the grass, with its perfect speckled eggs. He thought of the last of the apple blossoms covering the ground like a blanket of fragrant pink snow. He thought of his sheep and the three new lambs this spring. No, nothing was perfect, but neither was it ruined. It was a beautiful world, even with Themselves trying to spoil it. It was soil'd. It was a garden God walked in. John was sure of it.

## 6. Conclusions

We know that we are always influenced by others around us, sometimes unconsciously picking up on phrases or ideas or trends. In crass terms, that is the power of advertising, the fact that humans are suggestible and that we co-create our world. We can see that type of interplay, that conversation, that compost between the writers of the synoptic gospels, whose writing clearly was in conversation with one another. The Gospel of John seems to stand alone from this as a unique and artistic entity. While biblical performance criticism points to the Gospel's debt to ancient Greek drama as an explanation that it was not created ex nihilo, I also hope that, in this paper, I have made the case that the Gospel writer and the implied narrator—the beloved disciple—are in conversation not only with Jesus but also with the other exemplary characters of this Gospel, including with Mary of Bethany, who works with scent that inspires the evangelist's wordsmithing. Further, if Mary's actions anticipate Jesus's command in John 13, perhaps they also have an effect on Jesus, helping him discover a creative means to convey the prophetic and transformative message of love to this soil'd world.

**Funding:** This research received no external funding.

**Data Availability Statement:** The original contributions presented in the study are included in the article, further inquiries can be directed to the corresponding author.

**Conflicts of Interest:** The author declares no conflict of interest.

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
