# Peer review of "Mary of Bethany: Creation through Conversation"

_religions, doi:10.3390/rel15040411_

Round 1

Reviewer 1 Report

Comments and Suggestions for Authors

This is an interesting article especially in the context of the special issue. The discussion of the anointing in John 12 is strong. The transition from that discussion to the author's experience is rough. Instead of, "Here I turn..." maybe something like, "Now let's explore how my own experience reflects the sensory expression of this Gospel episode" or the like. The second part of the article could then more clearly link to the first part. This is currently not as clear and helpful as it could be.

Also, on p. 3, line 91, there is a citation of Mark that 100% should be a citation of John 12:7.

A typo: on that same page, line 103: "to which Martha referred to in the comment..." delete the second "to"

Author Response

 have written a new and longer introduction to the article that situates this article in the wider scholarly context as well as gives a clearer thesis statement. The concerns about performance art is better mapped out in the introduction.

The reviewer is correct that the two points (performance and aroma) were conflated. I approached this with a slightly different solution, moving some of the material on aroma into the introduction, and then in the discussion of performance, describing the nard/scent as the artistic medium in which Mary works. I think this is clearer without making aroma an entire separate point.

I appreciate the feedback about the transition between halves of the paper and the conclusion and I’ve tried to work with it. I do introduce a small new idea in the closing section (i.e. that the author of John is implied to be the beloved disciple and one of the characters). I’ve tried to both summarize the importance of the articles and to end once again on an artistic note that references the second section.

Reviewer 2 Report

Comments and Suggestions for Authors

For me the connection between the exegesis of John 12 and the artwork is not really clear. For the exegetical part: that could be more elaborated. Why is Mary's annointing somewhat like a artwork?

Author Response

I do not see notes from a second reviewer.

Round 2

Reviewer 2 Report

Comments and Suggestions for Authors

The revised version is ok now.

Author Response

Helpful feedback. The bolding was intended to show where I had made additions, given te instructions to make clear what had changed. It's all been removed and hopefully standardized now.